# Transcription Regulation of Anthocyanins and Proanthocyanidins Accumulation by Bagging in 'Ruby' Red Mango: An RNA-seq Study

Wencan Zhu [1,†], Hongxia Wu [2,†], Chengkun Yang [1], Xiaowen Wang [1], Bin Shi [1], Bin Zheng [2], Xiaowei Ma [2], Minjie Qian [1,*], Aiping Gao [3,*] and Kaibing Zhou [1,*]

1   Sanya Nanfan Research Institute & Key Laboratory of Quality Regulation of Tropical Horticultural Crop in Hainan Province, Hainan University, Haikou 570228, China; wencanzhu@hainanu.edu.cn (W.Z.); hndxyck@hainanu.edu.cn (C.Y.); xiaowenwang@hainanu.edu.cn (X.W.); shibin@hainanu.edu.cn (B.S.)
2   Key Laboratory of Tropical Fruit Biology, Ministry of Agriculture and Rural Affairs, South Subtropical Crops Research Institute, Chinese Academy of Tropical Agricultural Sciences, Zhanjiang 524013, China; wuhongxia@catas.cn (H.W.); zhengbin@catas.cn (B.Z.); maxiaowei428@126.com (X.M.)
3   Tropical Crops Genetic Resources Institute, Chinese Academy of Tropical Agricultural Sciences & Ministry of Agriculture Key Laboratory of Crop Gene Resources and Germplasm Enhancement in Southern China, Haikou 571100, China
*   Correspondence: minjie.qian@hainanu.edu.cn (M.Q.); aipingao@catas.cn (A.G.); zkb@hainanu.edu.cn (K.Z.)
†   These authors contributed equally to this work.

**Abstract:** The biosynthesis of anthocyanins and proanthocyanidins (PAs), components of two main flavonoids in plants, is regulated by environmental factors such as light. We previously found that bagging significantly repressed the biosynthesis of anthocyanins in red 'Ruby' mango fruit peel, but induced the accumulation of PAs. However, the molecular mechanism remains unclear. In the current study, transcriptome sequencing was used for screening the essential genes responsible for the opposite accumulation pattern of anthocyanins and PAs by bagging treatment. According to weighted gene co-expression network analysis (WGCNA), structural genes and transcription factors highly positively correlated to anthocyanins and PAs were identified. One flavanone 3-hydroxylase (*F3H*) and seven structural genes, including one chalcone synthase (*CHS*), one flavonoid 3'-hydroxylase (*F3'H*), one anthocyanidin synthesis (*ANS*), three leucoanthocyanidin reductase (*LARs*), and one UDP glucose: flavonoid 3-O-glucosyltransferase (*UFGT*), are crucial for anthocyanin and PA biosynthesis, respectively. In addition to MYB and bHLH, ERF, C2H2, HD-ZIP, and NAC are important transcription factors that participate in the regulation of anthocyanin and PA biosynthesis in 'Ruby' mango fruit peel by bagging treatment. Our results are helpful for revealing the transcription regulation mechanism of light-regulated mango anthocyanin and PA biosynthesis, developing new technologies for inducing flavonoid biosynthesis in mangos, and breeding mango cultivars containing high concentrations of flavonoids.

**Keywords:** *Mangifera indica*; flavonoids; transcriptome; light; weighted gene co-expression network analysis





## 1. Introduction

Flavonoids are secondary metabolites widely present in plants, known as small molecule compounds of polyphenols due to the presence of phenolic hydroxyl groups. Anthocyanins and proanthocyanidins (PAs) are two of the predominant subgroups of flavonoids in plants [1]. PAs, also called condensed tannins, are flavan-3-ols polymers, which can help plants resist biotic stress, such as preventing the invasion of pathogens and the harm caused by herbivores [2]. Anthocyanins, as one of the most important components of pigments in plants, are accumulated in various tissues and display bright colors, which can attract animals to help plants pollinate and spread seeds to different places [3,4]. In

addition, anthocyanins, as natural antioxidants in plants, can effectively prevent excessive damage to plants in various adverse conditions [5,6].

The biosynthesis of PAs and anthocyanins has been clarified in various plants, through the phenylpropanoid and flavonoid pathway [7]. The synthetic enzymes shared by PAs and anthocyanins include phenylalanine ammonia lyase (PAL), cinnamic acid 4-hydroxylase (C4H), 4-coumarate:CoA ligase (4CL), chalcone synthase (CHS), chalcone isomerase (CHI), flavanone 3-hydroxylase (F3H), flavonoid 3′-hydroxylase (F3′H), and dihydroflavonol 4-reductase (DFR). Leucoanthocyanidins, which are produced under the catalysis of DFR, can be further catalyzed to synthesize PAs by leucoanthocyanidin reductase (LAR), or anthocyanidins by anthocyanidin synthesis (ANS). Anthocyanidins cannot exist stably in plants, so they are quickly catalyzed by UDP glucose: flavonoid 3-*O*-glucosyltransferase (UFGT) or anthocyanidin reductase (ANR) to synthesize stable anthocyanins or PAs, respectively [8,9]. The expression of structural genes is regulated by the MYB-bHLH-WD40 (MBW) transcription factor (TF) complex, with the key function of R2R3-MYB [10].

Anthocyanin and PA biosynthesis is influenced by environmental conditions including light. Numerous studies on apples and grapes have demonstrated that coloration in the peel was promoted by light, and there is a significant difference in anthocyanin content between fruit peels exposed to sunlight and shaded [9,11]. Due to its great contribution to the fruit skin color improvement and protection against insects and pathogens, bagging treatment is widely used for pears [12], apples [13], and grapes [14]. Postharvest UV-B light supplementation can induce anthocyanins or PAs in fruit species such as apple [15], pear [12,16], grape [17] and mango [18]. In addition to ultraviolet light, blue light has also been found to promote anthocyanins but repress PA accumulation in mangos [19].

Mango (*Mangifera indica* L.) is a tropical fruit widely cultivated in both tropical and subtropical regions, with a yield of 54 million tons ranking them the fifth most cultivated fruit in the world (http://www.fao.org/faostat/, accessed on 10 January 2023). The skin color of mangos exhibit green, yellow, or red, and the red-colored cultivars are most preferred by the consumers. PAs, on the other hand, provide an astringent taste, which protect mango fruits against early consumption. At present, researches about the molecular mechanism of mango flavonoid biosynthesis mainly focus on the transcription changes of the flavonoid biosynthetic and regulatory genes [20–22], while how these genes are regulated by upstream genes has not been elucidated. In previous studies, we found that bagging the red mango cultivar 'Ruby' fruit significantly reduced the content of anthocyanins in the peel compared to unbagged fruits. However, surprisingly, the content of PAs in the peel was increased [23]. These results were divergent when compared with other studies, in which the biosynthesis of flavonoids, including anthocyanins and Pas, is usually induced by light [24–26]. We have only analyzed the expression of structural and regulatory genes that regulate anthocyanin biosynthesis [23,27]. Therefore, elucidating genes associated with anthocyanin and proanthocyanidin accumulation in mango peel by bagging treatment has both theoretical and applied significance.

In this study, the 'Ruby' mango peel of bagged and unbagged (control) fruit was collected at 50 days after full bloom (DAFB), 80 DAFB, and 120 DAFB, as described in the previous study [23]. Transcriptome sequencing was established to screen the candidate genes related to PA and anthocyanin accumulation regulated by light using weighted gene co-expression network analysis (WGCNA). Our results will enrich the theory of the regulation of fruit flavonoid biosynthesis by light.

## 2. Materials and Methods

### 2.1. Plant Materials and Treatments

The red mango cultivar 'Ruby' was used as the plant material and obtained from the South Asian Subtropical Research Institute (SSCRI) located in Zhanjiang, Guangdong Province, China, with a subtropical monsoon climate. The annual precipitation is 1396 to 1723 mm, and the annual average temperature is 22.7 to 23.5 °C. The treatment was conducted in the mango field genebank of the research institute. We selected three

healthy, disease- and pest-free fruit trees as three independent biological replicates. The trees were similar in size and number of fruits, and were subjected to uniform light conditions. The experiment was conducted in 2020. At 20 days after full bloom (DAFB), 50 fruits from each tree were bagged with double-layer black yellow paper bags (Kobayashi Co., Ltd., Qingdao, China), and the remaining unbagged fruits exposed to natural light were regarded as control. At 50, 80, and 120 DAFB, representing the early developmental stage, middle developmental stage and green mature stage, respectively, 10 fruits per tree from the treatment and control groups were harvested. The mango fruits were transferred to the laboratory, and a portable colorimeter (Linshang Technology Co., Ltd., Shenzhen, China) was used to measure the color index of the peel including *L** (lightness), *a** (red–green color), and *b** (yellow–blue color). Then the mango fruit peel was peeled off using a fruit peeler, quickly frozen in liquid nitrogen, and kept in an ultra-low temperature refrigerator at −80 °C.

## 2.2. RNA Extraction and Transcriptome Sequencing

RNA preparation pure plant reagent kit (Tiangen, DP441, Beijing, China) was used for total RNA extraction. cDNA was reverse-transcribed from the enriched and fragmented mRNA, which was further purified, repaired, and ligated to the adapter after A-tail addition. After the library construction, two end RNA sequencing (paired-end) was performed by Illumina sequencing platform (Metware Biotechnology Co., Ltd., Wuhan, China). Clean reads were obtained using Fastp software 0.20.0 to remove the low-quality data of the raw reads (https://github.com/OpenGene/fastp, accessed on 20 January 2023). Then, clean reads were mapped to the mango reference genome with the accession number of PRJCA002248 in the BIG Genome Sequence Archive database using TopHat [28]. Transcripts were assembled from reads by Cufflinks, and the level of gene expression was represented by Fragments Per Kilobase of transcripts per million fragments mapped (FPKM). The differential expression analysis between the two groups was completed using DESeq R package (1.10.1). Genes were considered as differentially expressed genes (DEGs), which have a significant *p*-value < 0.05 and $|\log_2 \text{FoldChange}| > 1$. The transcriptome raw data were submitted to NCBI with the ID number of PRJNA905802.

## 2.3. cDNA Synthesis and Quantitative Real-Time PCR (Q-PCR)

cDNA was synthesized with a total of 1μg RNA using a HiScript IIQ RT SuperMix (Vazyme, R223-01, Nanjing, China) kit. Q-PCR assay was performed according to the description by Shi et al. All Q-PCR primers are listed in Supplementary File Table S1, which were designed by the online website primer3 (https://bioinfo.ut.ee/primer3-0.4.0/, accessed on 10 March 2023). The relative expression level of the genes was calculated by the $2^{-\Delta\Delta Ct}$ method, and normalized by the mango *β-actin* gene.

## 2.4. WGCNA and KEGG Analysis

WGCNA and Kyoto Encyclopedia of Genes and Genomes (KEGG) analyses were carried out via online website cloud tools (https://cloud.metware.cn/, accessed on 5 March 2023). WGCNA analysis was performed using PA and anthocyanin concentrations in bagged and unbagged mango fruit peel at three different developmental stages and all the expressed genes obtained by RNA-seq. MergeCutHeight and minModuleSize were set to 0.15 and 30, respectively. Automatic network construction function 'blockwise' was used for module building. The eigengene value of each module was calculated to test the correlation between each sample and trait. Candidate genes from the 'red' and 'turquoise' modules were selected for subsequent analysis, with a threshold of 0.80. KEGG analysis was performed with candidate genes from the 'red' and 'turquoise' modules.

## 2.5. Statistical Analysis

The data were represented by mean value ± standard deviation. The color index data of the fruit peel were analyzed by Student's *t*-test using SPSS 27.0 (SPSS, Chicago,

IL, USA) to evaluate the statistical difference. Probability values of <0.05 (*) and <0.01 (**) were considered statistically significant and highly statistically significant, respectively. All the heatmaps were drawn by TBtools [29], and z-score standardization function was used between each row of data. The stacked bar chart was drawn by ChiPlot (https://www.chiplot.online/, accessed on 26 March 2023).

## 3. Results

### 3.1. Analysis of Fruit Color

The fruit color index $L^*$ reflects brightness, $a^*$ represents green ($-$) or red ($+$), and $b^*$ represents blue ($-$) or yellow ($+$), respectively. Compared to bagged 'Ruby' fruits, unbagged 'Ruby' fruit peel exhibited lower $L^*$, higher $a^*$, and lower $b^*$ at all developmental stages, with the skin consistently showing a deep red color (Figure 1) [23]. Interestingly, bagging treatment resulted in the 'Ruby' fruit peel showing a bright yellow color at 50 and 80 DAFB (Figure 1). However, at 120 DAFB, the fruit peel ultimately showed a light red color, with decreased $L^*$ and $b^*$, and increased $a^*$ (Figure 1).

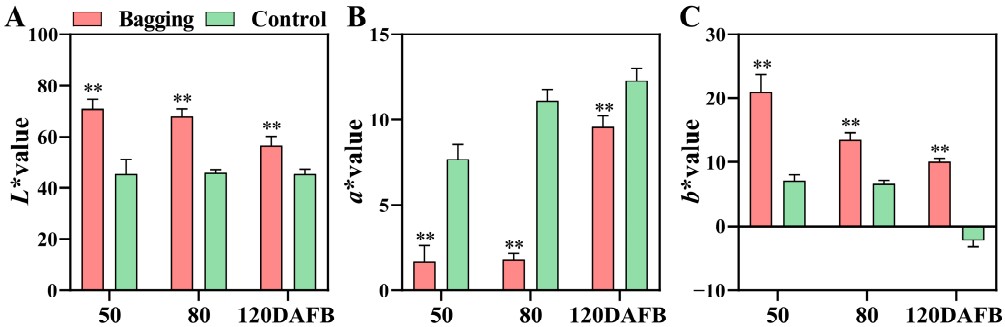

**Figure 1.** Effects of bagging and control (non-bagging) treatments on the color index $L^*$ (**A**), $a^*$ (**B**), and $b^*$ (**C**) values of mango fruit 'Ruby' at different developmental stages. Each value represents the mean $\pm$ standard deviation of three biological replicates. Two asterisks (**) are used to indicate the probability values of <0.01, according to Student's *t*-test results.

### 3.2. Transcriptomee Sequencing Data Overview

Samples for RNA extraction and sequence analysis were taken from bagged or exposed to natural light mango peel, at 50, 80, and 120 DAFB. There were 45.24–89.50 million (M) raw reads generated from each library of high-throughput sequencing (Table 1). After filtering the raw reads, 43.21–85.38 M clean reads were gained, including 6.48–12.81 G clean base (Table 1). Mapping the reads to the mango genome database resulted in 39.64–78.12 M mapped reads and 38.11–74.86 M unique mapped reads (Table 1). The percentage of sequencing errors, Q20, Q30, and GC content were approximately 0.02%, 98.44%, 95.21%, and 43.66%, respectively (Table 1).

**Table 1.** Transcriptome sequencing data overview.

| Classification | Maximum | Minimum | Average |
|---|---|---|---|
| Raw Reads | 89,500,794 | 45,244,014 | 67,372,404 |
| Clean Reads | 85,383,796 | 43,218,500 | 64,301,148 |
| Clean Base (G) | 12.81 | 6.48 | 9.645 |
| Mapped Reads | 78,126,639 | 39,641,845 | 58,884,242 |
| Unique Mapped Reads | 74,863,224 | 38,112,858 | 56,488,041 |
| Error Rate (%) | 0.02 | 0.02 | 0.02 |
| Q20 (%) | 98.73 | 98.16 | 98.445 |
| Q30 (%) | 95.91 | 94.51 | 95.21 |
| GC Content (%) | 43.97 | 43.35 | 43.66 |

### 3.3. qPCR Validation of Differentially Expressed Genes

To ensure the accuracy and reliability of transcriptome data, we randomly chose six DEGs for qPCR validation. The results showed a high correlation between transcriptome results and candidate gene expression patterns, and the correlation coefficient between the two approaches reached 0.809 (Figure 2A,B).

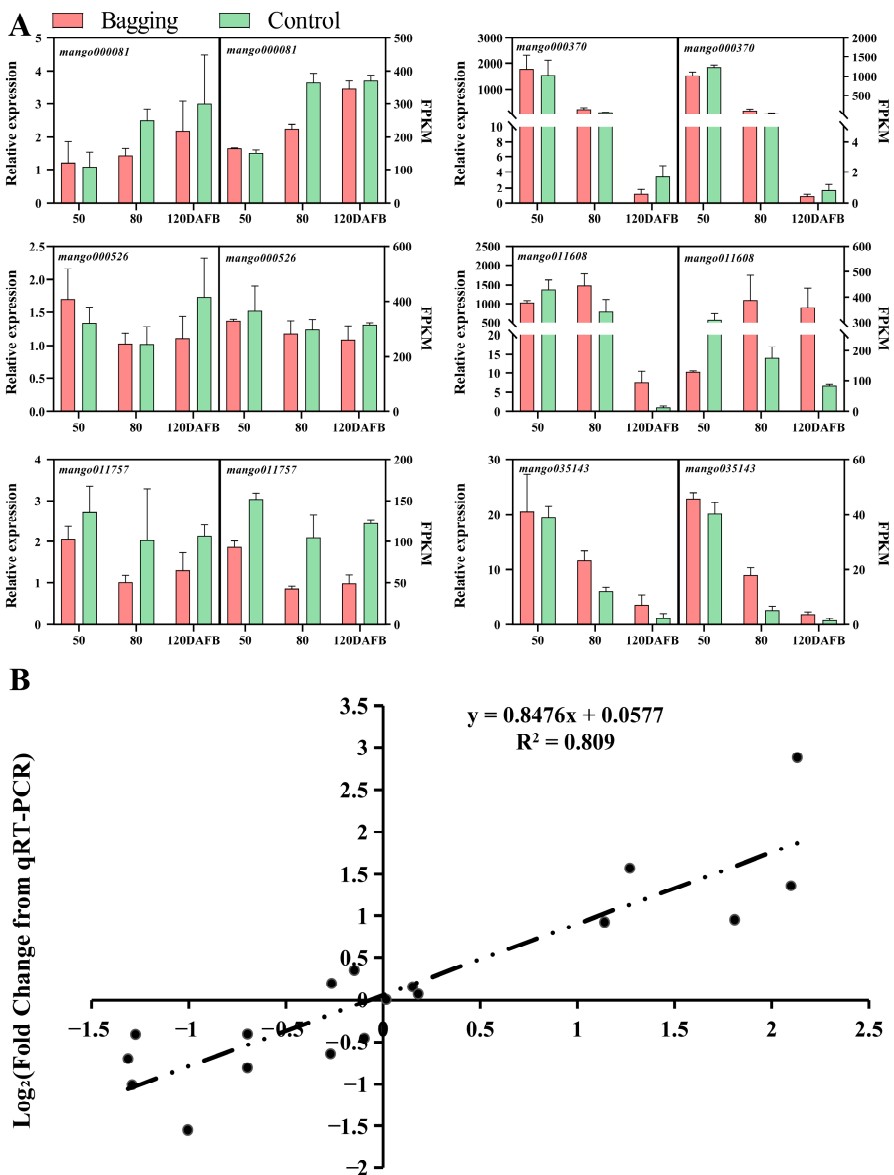

**Figure 2.** Verifying the transcripts of DEGs through qPCR. (**A**) qPCR and RNA-seq data of six randomly selected DEGs. The left side of each graph shows the relative gene expression levels obtained through qPCR analysis, while the right side shows the results based on RNA-seq. (**B**) Correlation analysis according to qPCR and transcriptome data results.

### 3.4. WGCNA Revealed PAs-Related and Anthocyanins-Related DEGs

To identify transcripts associated with the biosynthesis of anthocyanin and PAs, we conducted weighted gene co-expression network analysis (WCGNA) including all DEGs, and identified a total of 27 WCGNA modules (Figure 3). For PAs, module trait relationship analysis showed a high positive correlation between the turquoise module and procyanidin B3 content ($r = 0.88$, $p = 1.5 \times 10^{-6}$) (Figure 3). For anthocyanins, cyanidin-3-*O*-galactoside was highly positively correlated with the red module ($r = 0.99$, $p = 4.9 \times 10^{-15}$) (Figure 3). Unfortunately, no module showed a high correlation with procyanidin B1 according to

the threshold (±0.8), and no module showed a high negative correlation with PA and anthocyanin contents (Figure 3). Finally, genes from the turquoise and red modules were identified as candidate genes positively involved in anthocyanin and PA accumulation in mango peel, respectively (Figure 3).

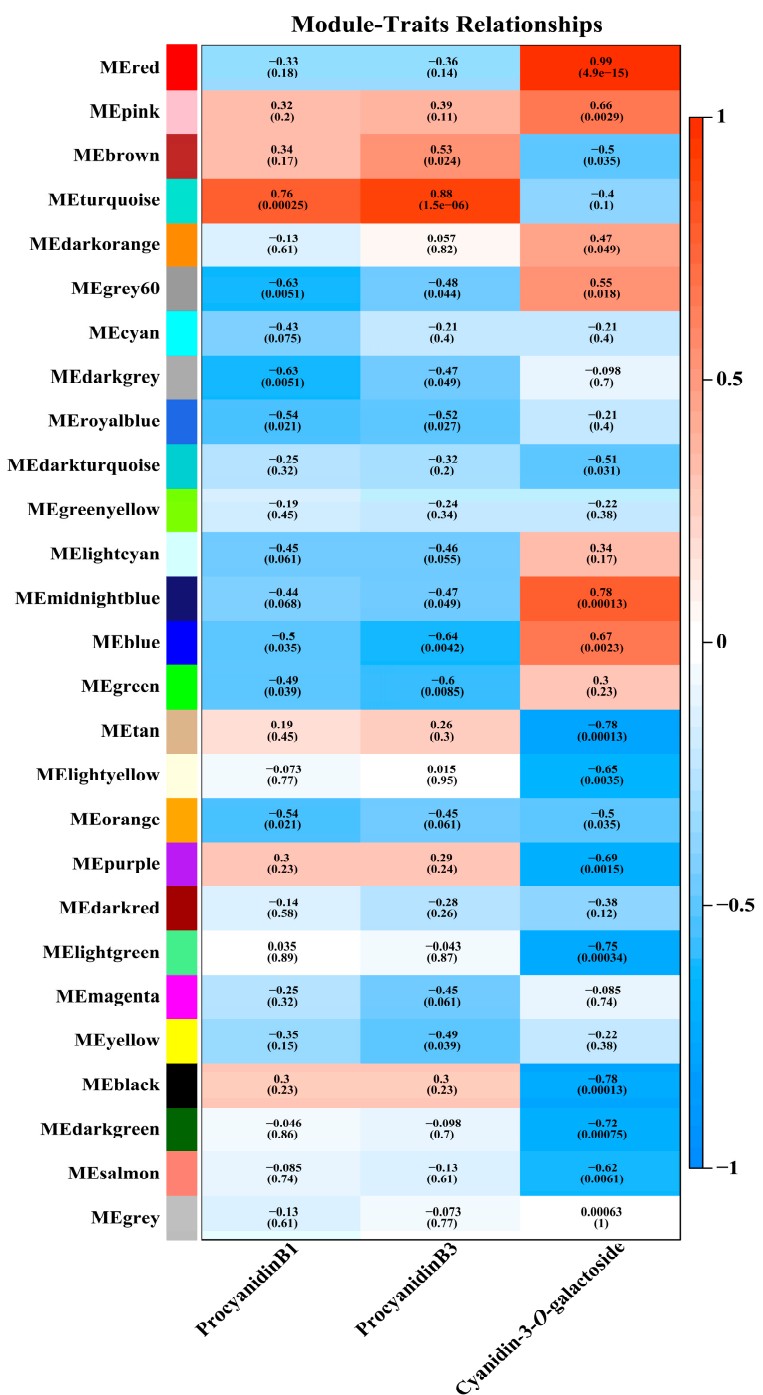

**Figure 3.** WCGNA of DEGs obtained from RNA-seq of bagged and non-bagged mango peel. The module feature correlation and corresponding *p*-values are shown in parentheses. There are 27 modules displayed on the left panel. The module trait correlation from −1 (blue) to 1 (red) is represented by the color change on the right. Procyanidin B1, procyanidin B3, and cyanidin-3-*O*-galactoside on the panel show changes in the concentration of corresponding substances.

### 3.5. Key Pathways Revealed by KEGG

Candidate genes from the turquoise and red modules identified by WGCNA were used for KEGG analysis. The results showed that most candidate module genes belonged to metabolic pathways (33.59% in turquoise, 49.43% in red) and biosynthesis of secondary metabolites (13.05% in turquoise, 28.62% in red) (Figure 4A,B). The turquoise module also enriched plant hormone signal transduction (10.75%) and plant–pathogen interaction (9.6%) (Figure 4A). In addition, carbon metabolism was enriched in red modules (8.46%) (Figure 4B).

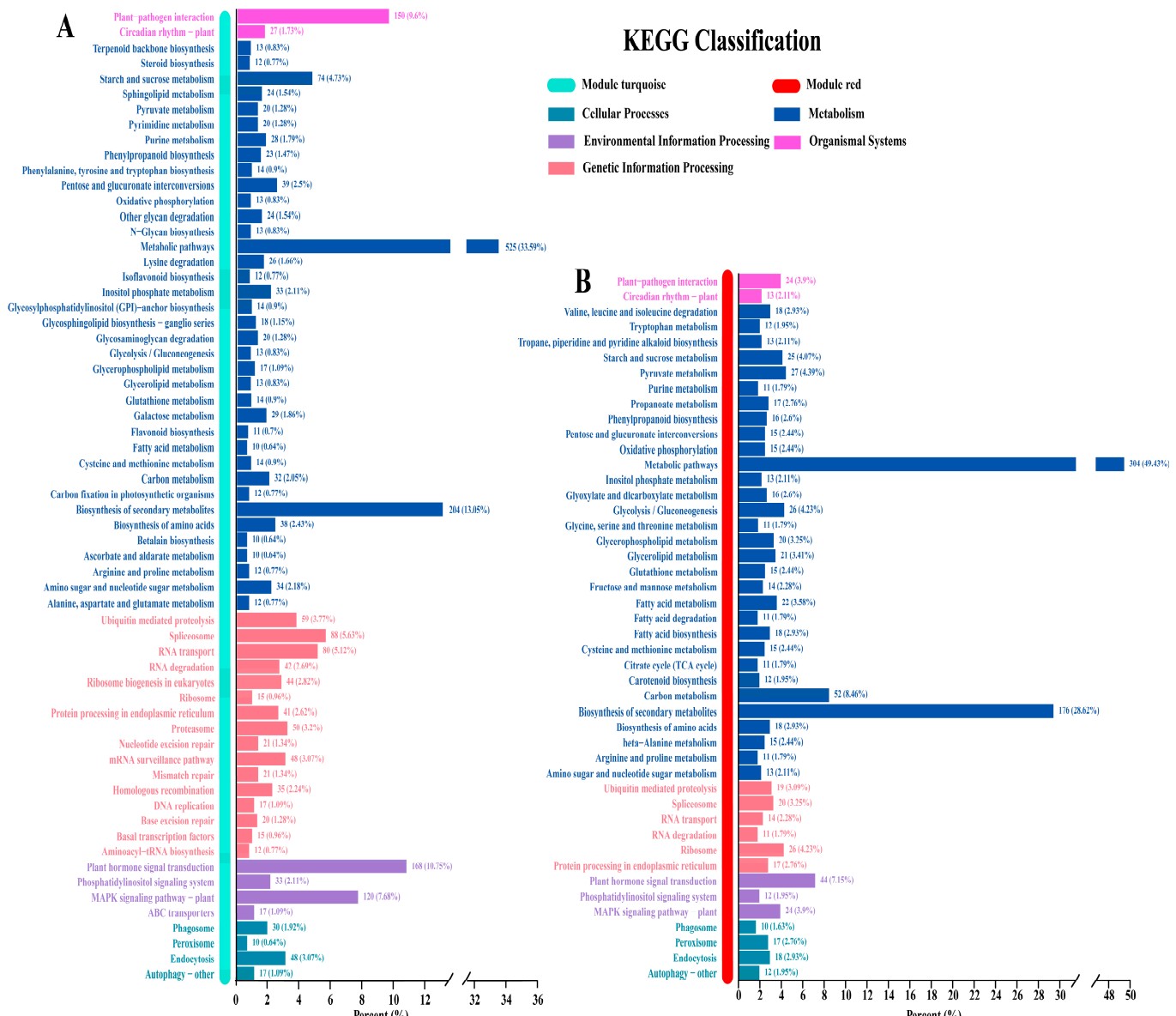

**Figure 4.** KEGG pathway enrichment analysis of transcripts from the turquoise (**A**) and red (**B**) modules associated with PA and anthocyanin biosynthesis.

### 3.6. Identification of Anthocyanin and PA Biosynthetic Genes by WGCNA

Based on WGCNA, a total of eight structural genes for anthocyanin and PA biosynthesis were identified (Figure 5). One *F3H* was positively correlated with the content of cyanidin-3-*O*-galactoside. One *CHS*, one *F3'H*, one *ANS*, three *LARs*, and one *UFGT* showed positive correlations with the concentration of procyanidin B3 (Figure 5). All structural genes which exhibited positive correlations with the concentration of procyanidin B3 were significantly induced in the bagged mango peel at 50 DAFB, while structural gene

positively correlated to the concentration of cyanidin-3-*O*-galactide was highly expressed in the control mango peel (Figure 5).

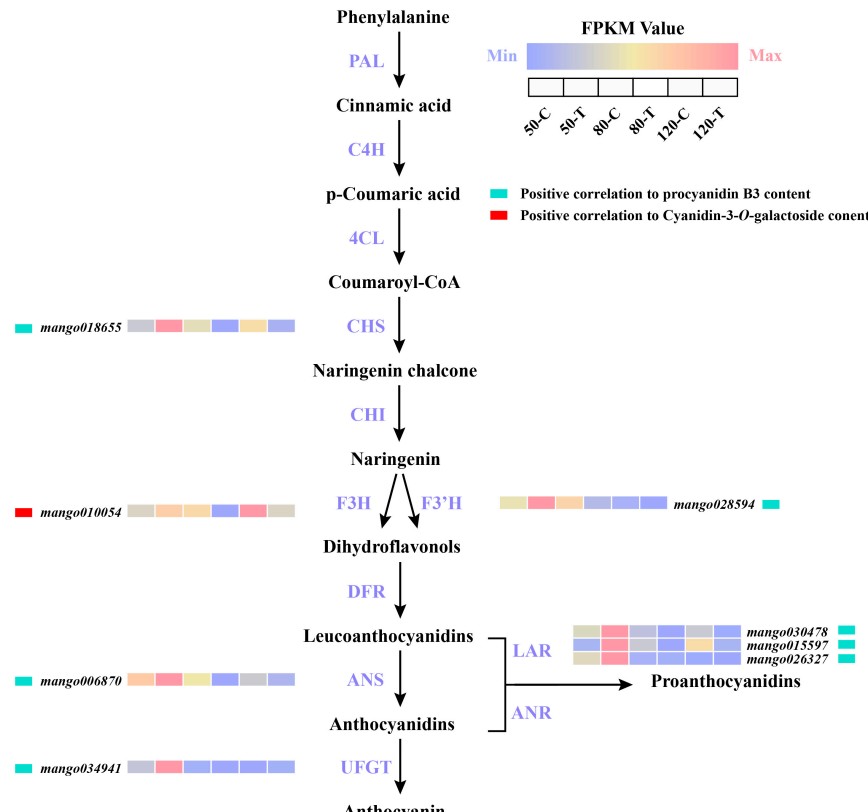

**Figure 5.** Transcription of structural genes related to PA and anthocyanin synthesis in mango fruit peel during three developmental stages by bagging and control (non-bagging) treatments based on WGCNA. The FPKM value from low to high is represented by color code from blue to red.

### 3.7. Identification of Transcription Factors by WGCNA

In total, eight *MYBs* related to the biosynthesis of PAs and anthocyanins were identified. Three of the members were positively correlated with the content of cyanidin-3-*O*-galactide, while five members were positively correlated with the content of procyanidin B3 (Figure 6A).

Regarding *bHLHs*, a total of 26 members were identified. Three members showed positive correlations with the concentration of cyanidin-3-*O*-galactide, and twenty-three members showed positive correlation with the concentration of procyanidin B3 (Figure 6A).

In addition to MYB and bHLH TFs, there were a total of 57 transcription factor gene families regulating PA and anthocyanin synthesis (Figure 6A). Among them, the *ERF* gene family has the largest number of members, with four members positively related to cyanidin-3-*O*-galactoside biosynthesis and twenty-three members positively related to procyanidin B3 content. *C2H2* took second place, and *HD-ZIP* and *NAC* ranked third place, with 19, 16, and 16 members identified as related to the synthesis of Pas and anthocyanins, respectively (Figure 6A). Moreover, more than 10 members of *NAC*, *Dof*, *ARF*, *WRKY*, *SBP*, and *MYB-related* regulatory gene were identified (Figure 6A). Figure 6B shows that the gene members positively correlated with the biosynthesis of procyanidin B3 exhibited the highest expression levels at 50 DAFB in bagged fruits. The genes from the red module showed generally higher expression levels in the unbagged fruit peel (Figure 6B).

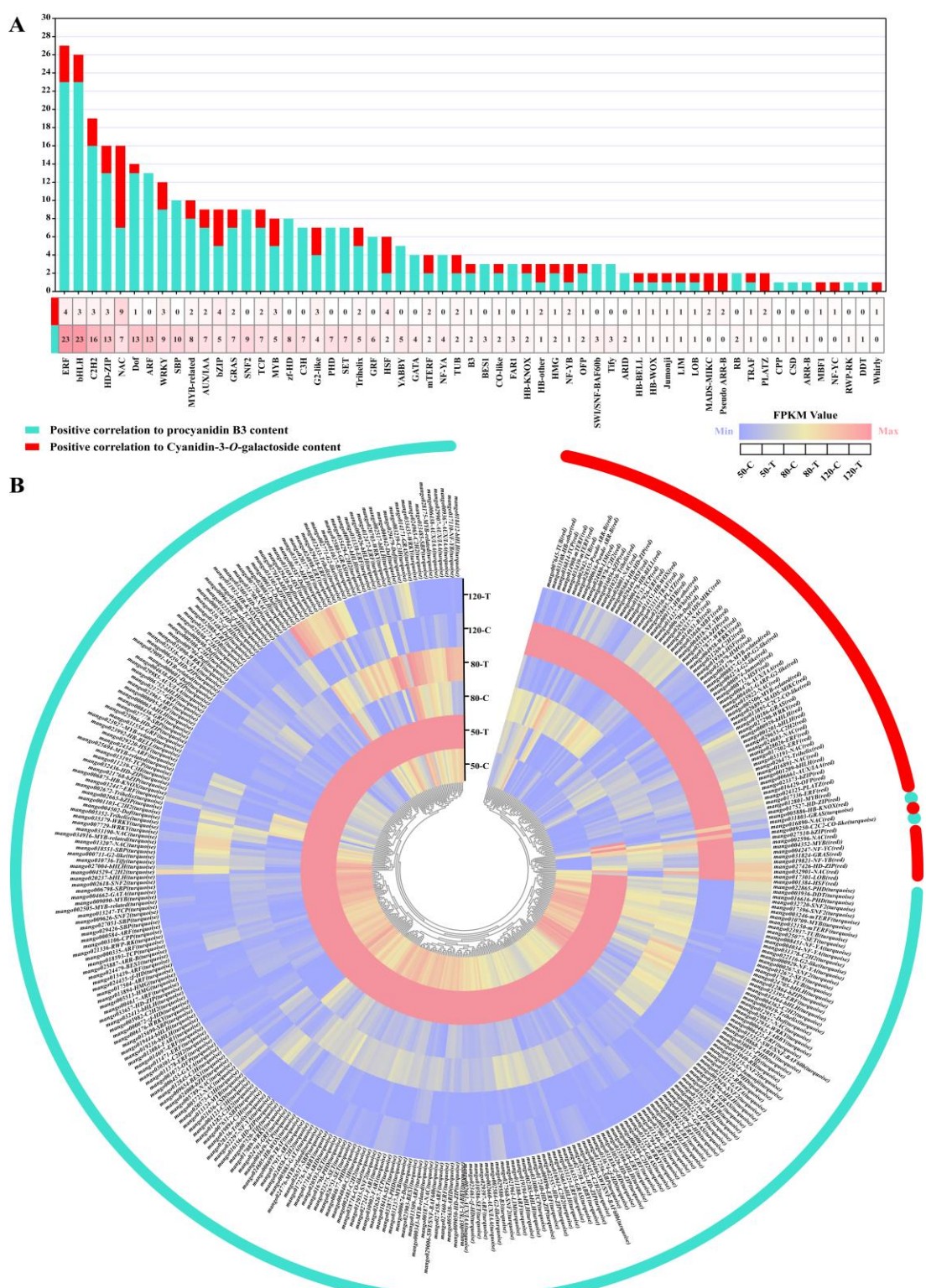

**Figure 6.** Transcription factor analysis involved in regulation of the biosynthesis of PAs and anthocyanins in mango peel. (**A**) The number of genes involved in PAs and anthocyanins in different transcription factor families. (**B**) The heatmap shows the expression patterns of transcription factors related to regulating the biosynthesis of PAs and anthocyanins in mango peel under bagging and control (non-bagging) treatments at 50, 80, and 120 DAFB.

## 4. Discussion

Light is one of the most essential environmental factors for plant growth and development. It is the energy source for plant photosynthesis, and serves as a signal to provide environmental information for plants. A large number of studies on pre-harvest and post-harvest light treatments have shown that light can regulate the biosynthesis of flavonoids in horticultural crops, for instance, cucumbers and mangos [6,18]. In our study, light induced the biosynthesis of anthocyanins in mango peel, which was consistent with other research results [30,31]. However, bagged fruit peel showed a higher content of PAs compared to unbagged fruit peel, which was opposite from other studies. Other studies have shown that bagging treatment can significantly inhibit the content of PAs [32,33]. Hence, revealing the mechanism related to light-induced biosynthesis of anthocyanins but repressed biosynthesis of PAs in mangos is meaningful.

A total of eight structural genes of anthocyanins and PAs synthesis pathways were identified in WGCNA analysis. Among them, bagging treatment resulted in significant higher expression of the three key structural *LAR* genes for synthesizing PAs, and the gene expression trend was consistent with changes in PA content in previous studies [23] (Figure 5). LAR, as a key enzyme that catalyzes the biosynthesis of PAs in the final step, has been shown to be positively correlated with PA content in strawberries and apples [34,35]. The *F3H* gene, which was positively correlated with anthocyanins content, showed up-regulated transcription in unbagged fruit peel during the three periods (Figure 5). *F3H* has been reported to contribute to anthocyanin synthesis in barley [36], and mulberries [37]. These results indicate that light has an inducing or inhibitory effect on different structural genes. In most reports, flavonoid biosynthesis structural genes are up-regulated by light [38,39], but there are also reports confirming that some structural genes are suppressed by light [12]. These results indicate that some flavonoids synthetic genes are also expressed at high levels under dark conditions to maintain the necessary flavonoid content levels to protect fruit against various stresses.

The MBW complex composed of R2R3-MYB, bHLH, and WD40 TFs is a key protein complex regulating flavonoid biosynthesis in plants. The MBW complex composed of TT2 (AtMYB123), TT8 (AtbHLH42), and TTG1 (WD40) in Arabidopsis can bind to the structural *ANR* and *UFGT* gene promoters to regulate PA and anthocyanin synthesis [40]. Coincidentally, homologous genes *FaMYB9/FaMYB11*, *FabHLH11*, and *FaTTG1* in strawberries, as well as homologous genes *GmTT2A*, *GmTT2B*, *GmTT8*, and *GmWD40* in soybeans, have also been reported to induce PA synthesis [34,41]. *MdMYB9* and *MdMYB11* [42], and *MdMYB12* [35] in apples have been found to bind to bHLH3 to induce the synthesis of PAs. In pears, light can induce the transcription of *bHLH64*, which can up-regulate the transcription of anthocyanin biosynthetic genes via interaction with MYB10 protein [43]. In this study, three *MYBs* and three *bHLHs*, and five *MYBs* and twenty-three *bHLHs* were positively correlated with cyanidin-3-*O*-galactoside and procyanidin B3 contents, respectively (Figure 6A). These results indicate that the regulation of anthocyanin biosynthesis is predominantly accomplished by a few MYB and bHLH TFs, while more MYBs and bHLHs are involved in the regulation of PAs accumulation.

In addition to MYB and bHLH, other TFs were also identified by WGCNA to regulate anthocyanin and PA biosynthesis. Among them, ERF, C2H2, HD-ZIP, and NAC showed the highest number of identified members (Figure 6A). ERF widely participates in regulating anthocyanin and PA biosynthesis. In strawberries, FaRAV1 promotes the transcription of *FaMYB10* and anthocyanin biosynthetic genes to induce anthocyanin biosynthesis [44]. In addition, PpERF24 and PpERF96 in pears respond to blue light signal and synergistically regulate anthocyanins synthesis by interacting with PpMYB114 [45]. The cysteine2/histidine2-type (C2H2) transcription factor family, as one of the most common subfamilies of Zinc-finger proteins [46], plays a positive role in anthocyanin biosynthesis in Arabidopsis and apples [47]. The Arabidopsis *AtANL2* is the first identified *HB* gene involved in anthocyanin synthesis belonging to HD-Zip group VI [48]. Four *HD-ZIP* genes were up-regulated during the low temperature-induced anthocyanins accumulation in zoysiagrasses (*Zoysia japonica*) [49]. NAC is also a large family of plant-specific transcrip-

tion factors, which has been widely reported to regulate flavonoid biosynthesis in plants. In the blood-fleshed peach, PpNAC1 promotes anthocyanin synthesis by activating the expression of *PpMYB10.1*, leading to the red flesh coloration [50]. MdNAC52 is involved in the biosynthesis of anthocyanins and PAs in apples by regulating *MdMYB9* and *MdMYB11*, respectively [51]. All these results suggest ERF, C2H2, HD-ZIP and NAC play crucial roles in regulating anthocyanins and PAs in mango, with the divergent response patterns to light condition among family members.

## 5. Conclusions

Compared with non-bagged mango fruits with dark red skin during the whole developmental stages, bagged mango fruits exhibit higher *L**, lower *a**, and higher *b** values in the color index, ultimately exhibiting light red color at 120 DAFB. Through RNA-seq and WGCNA analysis, genes derived from the turquoise module and the red module were identified to participate in the PA and anthocyanin synthesis in mango peel, respectively. KEGG analysis exhibited that most genes belonged to metabolic pathways and biosynthesis of secondary metabolites. In addition, eight transcripts of structural genes participating in the biosynthesis of anthocyanins and PAs in mango peel regulated by light were identified, as well as TFs including MYB, bHLH, ERF, C2H2, HD-ZIP, and NAC. Our research results provide a broad perspective of light-regulated anthocyanin and PA biosynthesis in mango, which is helpful for developing new technology to promote flavonoid accumulation in mangos and breeding new cultivars with high flavonoid content.

**Supplementary Materials:** The following supporting information can be downloaded at: https://www.mdpi.com/article/10.3390/horticulturae9080870/s1. Table S1: Sequences of the Q-PCR primers.

**Author Contributions:** Conceptualization, W.Z., H.W., M.Q., A.G. and K.Z.; methodology, W.Z., H.W., C.Y., X.W., B.S., B.Z. and X.M.; data curation, W.Z., H.W., C.Y., X.W., B.S., B.Z. and X.M.; writing—original draft preparation, W.Z., H.W., C.Y., X.W., B.S., B.Z., X.M., M.Q., A.G. and K.Z.; writing—review and editing, W.Z., H.W., C.Y., X.W., B.S., B.Z., X.M., M.Q., A.G. and K.Z.; funding acquisition, M.Q., A.G. and K.Z. All authors have read and agreed to the published version of the manuscript.

**Funding:** This research was funded by the National Natural Science Foundation of China (grant number: 32160678), the Major Science and Technology Plan of Hainan Province (grant number: ZDKJ2021014), the Hainan Provincial Natural Science Foundation of China (grant numbers: 322RC568; 320QN192), the Collaborative Innovation Center of Nanfan and High-Efficiency Tropical Agriculture, Hainan University (grant number: XTCX2022NYC04), the National Key Research and Development Plan of China (grant number: 2019YFD1000504), the earmarked fund from the China Agriculture Research System (grant number: CARS-31), Hainan Province Key Research and Development Plan (grant number: ZDYF2022XDNY255), and the Scientific Research Foundation of Hainan University (grant number: KYQD(ZR)20053).

**Data Availability Statement:** The transcriptome raw data were submitted to NCBI with the following ID number: PRJNA905802.

**Conflicts of Interest:** The authors declare no conflict of interest.

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
