# Peer review of "Transcription Regulation of Anthocyanins and Proanthocyanidins Accumulation by Bagging in ‘Ruby’ Red Mango: An RNA-seq Study"

_horticulturae, doi:10.3390/horticulturae9080870_

Round 1

Reviewer 1 Report

I have carefully reviewed the manuscript titled "Transcription Regulation of Anthocyanin and Proanthocyanidin Accumulation by Bagging in 'Ruby' Red Mango: RNA-seq Study" submitted to Horticulturae Journal. Overall, the paper presents a valuable contribution to the field of horticulture. However, I recommend addressing the following concerns before considering it for publication:

 Title: The current title is lengthy and could be revised to be more concise and impactful. A potential alternative could be "Transcription Regulation of Anthocyanin and Proanthocyanidin Accumulation by Bagging in 'Ruby' Red Mango: An RNA-seq Study." I suggest exploring further options for a shorter and more effective title.

 Importance and Objectives: The paper addresses an important aspect of red mango, an economically significant fruit species, by using RNA-Seq analysis to elucidate genes associated with anthocyanin and proanthocyanidin accumulation through bagging treatment. The significance of this research objective should be emphasized more explicitly in the manuscript.

 Language: The language of the manuscript is well-written and fluent. The authors have effectively conveyed their research findings.

 Introduction: In the introduction section, it is recommended to provide more detailed information on bagging applications by citing relevant literature. Specifically, the rationale, purpose, and benefits of bagging should be clearly explained, including its specific role in this study.

 Methods and Analyses: The methods and analyses have been appropriately planned and described. However, it is not clear from the manuscript why three different sampling days were chosen and what criteria guided their selection. It is crucial to provide a clear explanation for this choice, supported by scientific rationale.

 Alignment with Journal's Concept: The manuscript aligns well with the concept of Horticulturae Journal, focusing on the transcriptional regulation of anthocyanin and proanthocyanidin accumulation in 'Ruby' red mango through bagging treatment.

 Conclusion: The conclusion section should be further expanded to provide additional details on the potential benefits of this study for future research endeavors. A few sentences highlighting the implications of the findings for future studies would enhance the overall conclusion.

 In conclusion, I find this manuscript to be valuable and deserving of publication in Horticulturae Journal, provided that the aforementioned concerns are adequately addressed. The suggested improvements will enhance the clarity, scientific impact, and overall quality of the paper.

Reviewer 2 Report

Review report for the article

Horticulturae -2484505

The manuscript entitles “RNA-seq reveals the transcription regulation of the opposite accumulation pattern of anthocyanins and proanthocyanidins by bagging treatment in red mango cultivar ‘Ruby’” has been written well and have some comments below:

In abstract Line no 25: pls clear what is F3H and LARs? Please write full name first time use.

In line no 26, Pls clear the sentence and rephrase it with full form of and the output of those genes.

In general, authors need to improve the abstract. Also put conclusion and output from this study in brief.

In Keywords section, Keywords must be informative and different from the title.

In Introduction part, Authors need to rephrase the introduction part as it is very huge and big, so kindly reduce the length for intro and also reduce the number of references with recent one.  

Line no 104; What do you mean about DAFB? Please write full name first time use.

Line no 110, What is L*, a* and b*.?

Figure 1 should be more clearly and with a high resolution.

Figure 3 and 4 should be more clearly and with a high resolution.

In discussion also, reduce the number of references.

References should be according to the journals guideline. And follow same format for all the references. English language must be improved in throughout the manuscript.

English is good but need to improvement.

Reviewer 3 Report

The experience is an extension of previous research. Methodology unacceptable experience, too few plants from which samples were taken, (three), whether there were repetitions or not. We do not know in which year the study was carried out, how the plants were cared for during growth, in what climatic conditions they grew. Ten fruits as a sample from one treatment is not enough, especially since the experiment was not complicated. The fact that interesting genetic analyzes were performed does not justify this. The journal is in the field of horticulture so you can't disregard the methodology in terms of growing plants in the experiment.

Sufficient English language.
